

# The effects of Pilates vs. aerobic training on cardiorespiratory fitness, isokinetic muscular strength, body composition, and functional tasks outcomes for individuals who are overweight/obese: a clinical trial

Angeles Bonal Rosell Rayes[1], Claudio Andre B. de Lira[2], Ricardo B. Viana[2], Ana A. Benedito-Silva[3], Rodrigo L. Vancini[4], Naryana Mascarin[1] and Marilia S. Andrade[1]

[1] Departamento de Fisiologia, Universidade Federal de São Paulo, São Paulo, São Paulo, Brazil
[2] Faculdade de Educação Física e Dança, Universidade Federal de Goiás, Goiânia, Goiás, Brazil
[3] Escola de Artes, Ciências e Humanidades, Universidade de São Paulo, São Paulo, Brazil
[4] Centro de Educação Física e Desportos, Universidade Federal do Espírito Santo, Vitória, Espírito Santo, Brazil

Corresponding authors
Claudio Andre B. de Lira,
andre.claudio@gmail.com
Marilia S. Andrade,
marilia1707@gmail.com

## ABSTRACT

**Background.** Some studies have been conducted to verify the effects of *Pilates* for individuals who are obese, but conclusive results are not yet available due to methodological concerns. The present study aims to verify and compare the effects of Pilates and aerobic training on cardiorespiratory fitness, isokinetic muscular strength, body composition, and functional task outcomes for individuals who are overweight/obese.

**Methods.** Of the sixty participants, seventeen were allocated to the control group, since the intervention protocol (*Pilates* or walking sessions) was during their working hours. The remaining 44 participants were randomly allocated to one of two experimental groups (*Pilates* ($n = 22$)) or aerobic groups ($n = 21$). The *Pilates* and aerobic groups attended 60-min exercise sessions, three times per week for 8 weeks. The aerobic group performed walking training at a heart rate corresponding to the ventilatory threshold. The *Pilates* group performed exercises on the floor, resistance apparatus, and 1-kg dumbbells. The control group received no intervention. All volunteers were evaluated at the beginning and end of the intervention. The following assessments were conducted: food intake, cardiorespiratory maximal treadmill test, isokinetic strength testing, body composition and anthropometry, abdominal endurance test, trunk extensor endurance test, flexibility test and functional (stair and chair) tests.

**Results.** There was no significant difference pre- and post-intervention in calorie intake [$F(2, 57) = 0.02744$, $p = 0.97$)]. A significant improvement in oxygen uptake at ventilatory threshold ($p = 0.001$; $d = 0.60$), respiratory compensation point ($p = 0.01$; $d = 0.48$), and maximum effort ($p = 0.01$; $d = 0.33$) was observed only in the *Pilates* group. Isokinetic peak torque for knee flexor and extensor muscles did not change for any groups. Lean mass ($p = 0.0005$; $d = 0.19$) and fat mass ($p = 0.0001$; $d = 0.19$) improved only in the *Pilates* group. Waist and hip circumference measurements decreased similarly in both experimental groups. Abdominal test performance improved

more in the *Pilates* group ($p = 0.0001$; $d = 1.69$) than in the aerobic group ($p = 0.003$; $d = 0.95$). Trunk extensor endurance and flexibility improved only in the *Pilates* group ($p = 0.0003$; $d = 0.80$ and $p = 0.0001$; $d = 0.41$, respectively). The *Pilates* group showed greater improvement on the chair and stair tests ($p = 0.0001$; $d = 1.48$ and $p = 0.003$; $d = 0.78$, respectively) than the aerobic group ($p = 0.005$; $d = 0.75$ and $p = 0.05$; $d = 0.41$, respectively).

**Conclusion**. *Pilates* can be used as an alternative physical training method for individuals who are overweight or obese since it promotes significant effects in cardiorespiratory fitness, body composition, and performance on functional tests.

# INTRODUCTION

Obesity constitutes an important threat to global public health in terms of prevalence, incidence, and economic burden (*Tremmel et al., 2017*). According to the World Health Organization (WHO), more than 1.9 billion adults aged 18 years and older are overweight (*World Health Organization, 2015*); of these, over 650 million are obese, and this number is growing exponentially worldwide (*World Health Organization, 2015*). The most common cause of obesity identified in a recent review study was "combined physical inactivity and inadequate diet" (*Ross, Flynn & Pate, 2016*). In spite of obesity being associated with an increased risk of premature death (*Xiao & Fu, 2015*), in a recent review, *Barry et al. (2014)* concluded that unfit individuals with lower cardiorespiratory levels had twice the risk of mortality than fit individuals, regardless of body mass index (BMI). Although fatness is associated with several chronic diseases, data suggest that the low cardiorespiratory levels in obese people are worse for mortality risk than the fatness, (*Barry et al., 2014*), so physical activity programs for obese people should aim to improve physical fitness in addition to reducing total body mass.

There is no consensus in the literature as to whether physical activity is important for total body mass loss and improved body composition (*Pedersen & Saltin, 2015*). The amount of body mass loss associated with the introduction of physical activity probably is not very high, considering the much lower degree of energy imbalance created by increased physical activity than by food restriction (*Wing, 1999*). However, several studies showed that physical training promotes a decrease in fat mass and in abdominal fatness, unlike diet intervention, which, in a harmful side-effect of very restrictive calorie diets, also promotes a fat-free mass loss (*Ross & Janssen, 1999*).

Moderate walking or jogging programs have traditionally been suggested to improve variables related to health (*Ross et al., 2000*; *Pedersen & Saltin, 2015*), however, in a recent review, *Mabire et al. (2017)* demonstrated that the effects of the walking training depend on the sex, age and body mass index. For obese people it is also difficult to exercise with weight bearing activities without supervision, because obesity is an important risk factor
involved in knee pain and initiation and progression of osteoarthritis (*Santangelo et al., 2016*). Adherence levels to traditional aerobic programs are not very high, particularly if they are monotone and potentially painful for obese individuals (*Burgess et al., 2017*).

Alternative exercises should therefore be considered, such as planning enjoyable activities or amusing, non–weight-bearing exercises that consider the importance of improving aerobic fitness, muscular condition, and body composition. One option is *Pilates*, which is a body/mind training regime involving a variety of exercises for core stability, muscular strength, flexibility, attention to muscle control, posture, and breathing (*Penelope, 2002*; *Aladro-Gonzalvo et al., 2012*; *Küçükçakı, Altan & Korkmaz, 2013*; *de Oliveira Francisco, de Almeida Fagundes & Gorges, 2015*). Because of this variety, *Pilates* avoids monotony can be performed in a seated, standing, or lying position, thus protecting the knee joints and avoiding weight bearing. *Pilates* apparatus, such as the Reformer, are constructed in a manner that can accommodate many human anatomic variations and can be adjusted such that similar properties of movement sequencing can be applied to a variety of body types and limb/torso lengths (*Muscolino & Cipriani, 2004*), which is of fundamental importance for obese people.

Some studies have been conducted to verify the effects of *Pilates* for obese individuals (*Jago et al., 2006*; *Aladro-Gonzalvo et al., 2012*), but, due to methodological concerns (e.g., lack of control group, small sample size, lack of a complete methodological description), conclusive results are not yet available. More high-quality studies assessing different physical fitness parameters have been suggested to establish the effects of *Pilates* (*Aladro-Gonzalvo et al., 2012*).

This study compares the effects of *Pilates* and traditional aerobic training (walking) programs on body composition, cardiorespiratory fitness, muscular performance (strength, endurance, and flexibility), and functional tasks for overweight or obese subjects without changes in diet. We hypothesized that *Pilates* results in similar benefits in cardiorespiratory fitness and body composition and higher benefits in isokinetic muscular strength and functional task outcomes compared to traditional aerobic training

## MATERIALS AND METHODS

### Participants

Female and male adults ranging from 30 to 66 years of age who are overweight or obese were invited to participate in the study. The medical school where the study was performed included almost 5,500 staff members and 6,000 students, and the volunteers were drawn from this community. Inclusion criteria were: present BMI higher than 25 kg/m$^2$ and be classified as insufficiently active or sedentary according to International Physical Activity Questionnaire (IPAQ) (*Craig et al., 2003*). Exclusion criteria were present neurological, cognitive, orthopedic and respiratory disease. Subjects presenting cardiovascular or endocrine disease which were not controlled were also excluded. In addition, subjects who presented cardiovascular events during the exercise tests were also excluded. From July 2013 until December 2014, 115 volunteers performed a medical exam taking into account a medical history questionnaire and a medical physical exam. The follow-up

for the first group of participants began in September 2013 and the last group ended in July 2015. Forty-six volunteers were excluded after the application of the inclusion and the exclusion criteria; the reasons for exclusion were: presence of cardiovascular disease ($n = 45$) and presence of arrhythmia ($n = 1$).

Of the 69 remaining participants, 25 were allocated to the control group, since the intervention protocols (*Pilates* or walking sessions) occurred during their working hours. The remaining 44 participants were randomly assigned by drawing their names written on pieces of paper from an opaque envelope and placing them in one of two experimental groups (*Pilates* or aerobic). During the entire intervention protocol, 9 volunteers dropped out of participation. One participant from the aerobic group withdrew due to knee pain. Eight participants from the control group did not attend the retest protocol; two of them reported back pain, two had undergone emergency surgeries (appendix and fracture), and four cited personal reasons. A flow diagram illustrating the study design and volunteer participation is presented in Fig. 1.

Comorbidities were detected during the medical exam, through self-reporting, with the following description: three participants with type 2 diabetes mellitus (*Pilates* group); 14 participants with hypertension (nine in the *Pilates* group, two in the aerobic group, and three in the control group); 18 participants with dyslipidemia (10 in the *Pilates* group, three in the aerobic group, and five in the control group), and 11 participants with hypothyroidism (7 in the *Pilates* group, two in the aerobic group, and two in the control group). All these presented comorbidities were under treatment and were controlled.

Subjects were informed of the study purpose and protocols, and all provided written informed consent before taking part in the study. The Human Research Ethics Committee of the University approved all experimental procedures (Approval number: 197.451) that complied with the principles outlined in the Declaration of Helsinki. This clinical trial was registered with the Brazilian Registry of Clinical Essays (ReBEC, *Registro Brasileiro de Ensaios Clínicos*—http://www.ensaiosclinicos.gov.br) and its registration code is RBR-7QNSH6. The authors confirm that all ongoing and related trials for this intervention were duly registered. It is noteworthy that the target sample size described in the clinical trial's registration was 20 participants for each group. To ensure that 20 participants would conclude the study, more than 20 subjects were initially recruited, and after withdrawals, 17 participants from the control group, 21 from the aerobic group and 22 from the *Pilates* group participated through to the conclusion of the study.

The age of the participants who completed all phases of experimental protocol ($n = 60$) was $55.9 \pm 6.6$yrs (*Pilates* group), $42.4 \pm 7.0$yrs (aerobic group) and $45.5 \pm 9.3$yrs (control group). One-way ANOVA (analysis of variance) was used to verify differences in age and height before intervention. The one way ANOVA revealed that Pilates group was older than the other groups [$F(2, 57) = 18.521, p < 0.01$]. The height was not significantly different between Pilates ($161 \pm 6$ cm), Aerobic ($161 \pm 8$ cm) or Control groups ($162 \pm 7$) [$F(2, 57) = 0.25089, p = 0.78$].
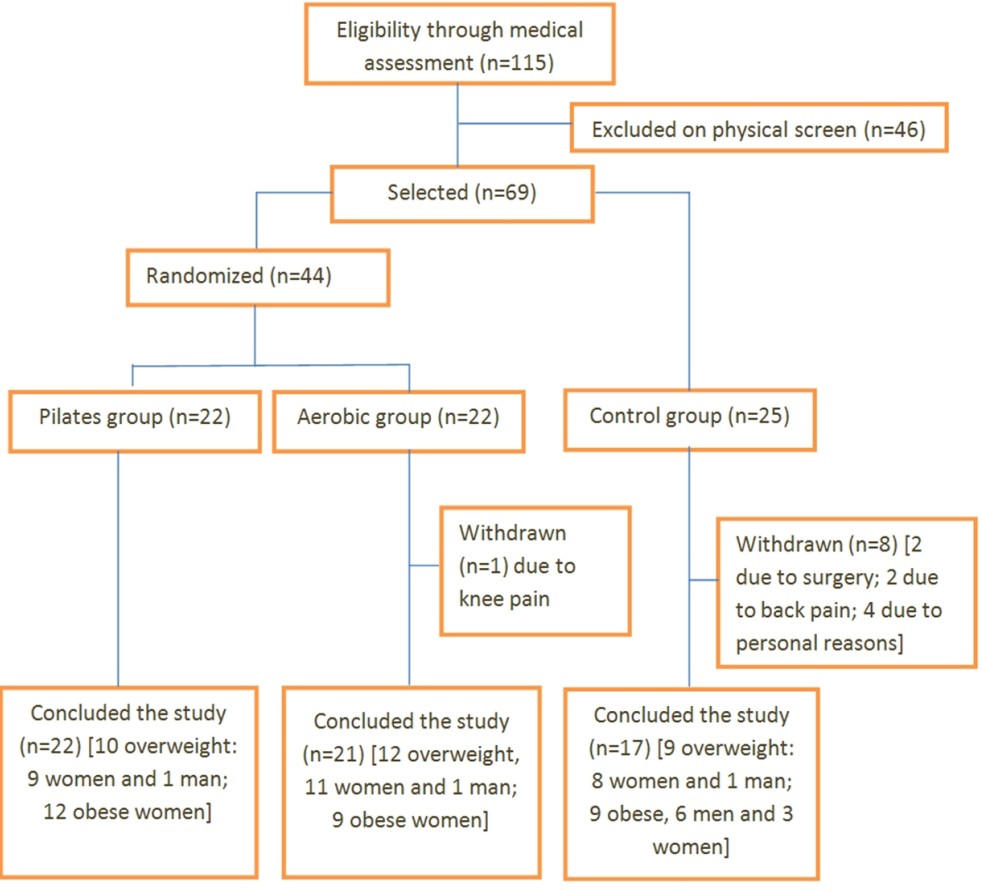

**Figure 1** Flowchart showing the Consolidated Standards of Reporting Trials (CONSORT) of participants through the study.

## Study design

This study was a variation of the classic RCT for experimental groups. According to *West et al. (2008)*, classical RCTs rely on the strong adherence of volunteers and, in some situations, very intense encouragement is necessary to guarantee the adherence, creating an unreal situation. In an alternative to this model, volunteers are allowed to choose the treatment they want to receive, rather than being randomly allocated in a group. This option is very interesting situations in which it is impractical or unrealistic to require adherence, and in which the adherence would only be possible through unrealistic encouragement. Considering that some participants could not participate in interventions due to their work activities, we decided to use a randomized encouragement design, since it would be unethical and unrealistic to request the volunteer to resign. Professionals who carried out the tests and measurements were blinded to the treatment conditions.

Since previous studies investigating aerobic training effects had found positive results with eight weeks of training (*Ho et al., 2012*; *Many et al., 2013*), participants in the aerobic and *Pilates* groups followed an experimental protocol for a period of eight weeks, with

60-minute sessions three times per week (totaling 24 sessions). Participants were asked not to initiate any other exercise activity apart from the one included in this study. When a participant missed three or fewer training sessions, the sessions were replaced at the end of the period, but if more than three sessions were missed, the participant was excluded from the study. Regarding adherence, only three participants needed to replace two exercise sessions at the end of the period, one from the *Pilates* group and two from the aerobic group. Therefore, by the end of the study, all participants for the aerobic and *Pilates* groups had completed 24 training sessions.

All participants from the three groups underwent tests performed over two consecutive days at the Exercise Physiology Laboratory of the Federal University of São Paulo (São Paulo, Brazil) both before and after the experimental protocol took place. The following assessments were conducted: food intake, cardiorespiratory maximal treadmill test, isokinetic strength testing, body composition and anthropometry, abdominal endurance test, trunk extensor endurance test, flexibility test and functional (stair and chair) tests.

As the control group did not follow an experimental protocol, for ethical reasons, control participants were offered the opportunity to participate in the *Pilates* or Aerobic program, according to the participant's interest, after the conclusion of the study. Those who participated in the experimental protocol were offered the opportunity to participate in the other protocol that they did not take part in initially. Details of the study protocol can be found at http://dx.doi.org/10.17504/protocols.io.p5wdq7e.

## Assessments
### *Food intake assessment*
Participants were asked to record their food intake for three days (two week days and one weekend day) (*Jeon et al., 2011*). The participants were asked to write down a description of the foods consumed, quantity per unit or portion sizes, type of preparation, and time of day. The quantitative assessment of the food record was carried out by calculating total energy consumption. An experienced nutritionist performed intake analysis using the Diet Win Professional 2.0 software (Brazil). Participants were instructed to avoid changing their diet habits during the experimental protocol.

### *Cardiorespiratory maximal treadmill test*
All participants submitted to a cardiorespiratory maximal test on a motorized treadmill (Inbrasport, ATL, Porto Alegre, Brazil) with individualized ramp protocol in order to identify VT, respiratory compensation point (RCP), and maximal or peak oxygen uptake ($\dot{V}O_2max/peak$). After one minute standing on the treadmill, the participants began the test with 3 min of warm-up at 3 km/h and 0 degree of inclination. From the 4th minute of testing, both speed (0.5 km/h) and inclination (0.5%) increased every 20 s. At the 6th minute of the test, the participant reached the final speed (6 km/h), while the inclination was increased every 20 s until the volunteer reached exhaustion. We chose this strategy because overweight/obese population traditionally do not reach high velocities in CPET and due to safety reasons. The same individual protocol for each volunteer was used in the retest.

Individual subjective level of exertion was obtained using the Borg scale (*Noble et al., 1983*). In order to determine VT, RCP, and ($\dot{V}O_2$max/peak), ventilation and expired gases were measured breath by breath using a metabolic analyzer (Quark, Cosmed, Italy), and all the measured data were considered as a mean of 20 s for analysis. Calibration procedures were performed according to the manufacturer's guidelines before each test. VT and RCP were determined separately by two different experienced investigators, according to established criteria (*Beaver, Wasserman & Whipp, 1986*). In cases where there was no consensus between the investigators, a third experienced investigator was asked to determine VT and RCP. VT was determined based on the following criteria: inflection in the ventilation curve, increase in the ventilatory equivalent for oxygen without increase in the ventilatory equivalent for dioxide carbon, and increase in partial pressure of exhaled oxygen. RCP was determined based on inflection in the ventilation curve, increase in the ventilatory equivalent for oxygen and in the ventilatory equivalent for dioxide carbon, and decrease in partial pressure of exhaled dioxide carbon (*Whipp, Ward & Wasserman, 1986*). Briefly, VT represents the most intense effort (expressed in terms of $\dot{V}O_2$ and/or workload) that can be performed without accumulation of lactate in blood (*Whipp, Ward & Wasserman, 1986*).

$\dot{V}O_2$max was the higher oxygen consumption value at the end of an incremental test, despite an increase in the treadmill speed, featuring the $\dot{V}O_2$ plateau. However, in clinical testing situations, a clear plateau may not be detected. Consequently, $\dot{V}O_2$peak is often used as an estimate for $\dot{V}O_2$max. For practical purposes, $\dot{V}O_2$max and $\dot{V}O_2$peak are used interchangeably. $\dot{V}O_2$max is a useful measure of exercise tolerance, and low values indicate poor aerobic physical fitness, and these data also have clinical significance (*Howley, 2007*).

### Internal training load

After each training session, the aerobic and *Pilates* groups were asked to provide a rating of perceived exertion (RPE, Borg scale) adapted by *Foster et al. (2001)*, to calculate the load of each session. Participants were asked about the rate of overall intensity of the training session 30 min after the end of the activity. The result was multiplied by the duration of the session, producing an index to monitor the internal training load.

### Isokinetic strength testing

Before the isokinetic testing, the participants performed a 5-min warm-up on a cycle ergometer (Cybex Inc., Ronkonkoma, NY, USA) at a resistance level of 25 watts, followed by low-intensity dynamic stretching exercises for the hamstrings and quadriceps to avoid stretching influence in strength values (*Mascarin et al., 2015*). Following the warm-up period, participants were placed on the isokinetic dynamometer (Biodex Medical Systems Inc., Shirley, NY, USA) to evaluate the isokinetic concentric strength of both lower limbs in a random order. Peak torque (PT) of knee flexor and extensor muscles (both dominant and non-dominant) in concentric activity was measured. Concentric activity was evaluated at 60°/s (the lowest speed in order to avoid high joint pressure while producing the highest torque values) and 240°/s (an angular speed closer to knee angular speed during the gait) separated by a one-minute rest (*Fleury et al., 2011*). Low angular speed (60°/s) was tested first. Participants completed three submaximal trials before each angular speed test to

familiarize themselves with the equipment, and five maximal repetitions to test at 60°/s and at 240°/s. The isokinetic dynamometer calibration procedure was performed according to the manufacturer's recommendations.

### Body composition and anthropometry

Body composition was assessed by dual-energy X-ray absorptiometry (DXA, software version 12.3, Lunar DPX, Madison, WI, USA) in order to measure fat-free mass and fat mass. This method has been previously demonstrated to be a safe method with minimum radiation dosage and reliable technique for fat mass (*Choi et al., 2015*). BMI was calculated by dividing body mass by height squared (kg/m$^2$). A BMI $\geq$ 25 kg/m$^2$ indicates overweight, while a BMI $\geq$ 30 kg/m$^2$ indicates obesity (*World Health Organization, 2012*).

Two different points of body circumference were considered: waist circumference (in cm) was measured around the midpoint between the lower margin of the last palpable rib and the top of the iliac crest; hip measurement was taken at the maximum circumference over the buttocks (*Reidpath et al., 2013*).

### Abdominal endurance test (sit-up test)

Abdominal endurance was evaluated using the partial trunk flexion test. Participants performed the test lying down with knees flexed at 90°, feet flat on the floor and hands clasped behind the head. The trunk was raised to the point where the shoulders were lifted from the mat. The maximum number of repetitions in one minute was measured (*Sarti et al., 1996*).

### Trunk extensor endurance test

Subjects underwent the Sorensen test to evaluate isometric endurance of the trunk extensor muscles. The participant lay prone on an examination table with the upper border of the iliac crests aligned with the edges of the table. The lower body was secured by three belts located around the hip, knees, and ankles. The arms were crossed over the chest. Participants were asked to isometrically maintain the upper body in a horizontal position as long as possible. The time to exhaustion was measured in seconds. The test stopped after a maximum of 240 s (*Demoulin et al., 2006*).

### Flexibility test

Flexibility was evaluated using the sit and reach test, which specifically measures the flexibility of the lower back and hamstring muscles. In this test, the participants sit on the floor with lower limbs fully extended, ankles flexed, and feet flat against the box used for the test. The participant must lean slowly and protrude forward as far as possible, sliding their fingers along a ruler affixed to the upper side of the box. After two practice reaches, the participant reaches out and holds the position for two seconds while the distance (in centimeters) is recorded (*Wells & Dillon, 1952*).

### Functional tests

Two different tests were performed to assess functional performance in activities of daily living: the stair test and chair test. In the stair test, the subject must climb up and down a set of 12 steps of 16 cm in height (total of 24 steps) twice. In the chair test, the subject must

get up out of and sit back down into a 50 cm-high chair 10 times. The runtimes for the two tests were recorded in seconds. These tests were adapted from *Pescatello et al. (2014)*.

## Interventions
### *Pilates protocol*
Participants who were in the *Pilates* group used a heart rate (HR) monitor (Suunto, Ambit, Finland) during all sessions to verify HR, in order to know at which percentage of maximal HR the volunteers performed the Pilates and in order to compare the mean HR obtained (exercise intensity) of the two experimental protocols. This intervention was performed at a *Pilates* Studio (CGPA *Pilates* Studio, São Paulo, Brazil) in groups (5 individuals per group), oriented by the same qualified professional who had graduated in sports and exercise science more than 20 years earlier and had experience in *Pilates* for 7 years. Each session was composed of: pre-*Pilates* warm-up exercises (10 min. breathing, shoulder, hip and spine alignment); *Pilates* (45 min. general conditioning with the traditional *Pilates* repertoire of exercises for beginners and intermediates, mat *Pilates,* and the following apparatus/equipment: reformer, cadillac, chair with adjustable spring resistance; high barrel, and accessories like magic circles and 1-kg dumbbells); and cooldown (5 min. respiratory exercises). All *Pilates* exercises were performed in a single series of repetitions, and the number of exercises was increased according to the participant's ability to successfully complete the exercise (Table 1). When necessary, exercises were adapted for obese and overweight body dimensions. An example of adaptation is in the Foot Work exercise on the Reformer. The original position of the exercise would be lying down on the Reformer, however some volunteers did not feel comfortable. A modification was made to elevate the trunk to the recline position, with the help of a wooden board leaned against the shoulder guard. In this way the volunteers could do the exercise comfortably, with no suffering.

### *Aerobic training protocol*
After cardiorespiratory maximal test (see details below), ventilatory threshold (VT) was determined and each volunteer performed the walking program in the HR corresponding to the VT. This walking intensity is considered safe to prevent orthopedic injuries in participants with obesity (*McQueen, 2009*). Each section of the program lasted 60 min and was performed three times per week for 8 weeks. In the first 10 min of each section, volunteers performed warm-up exercises. After the warm-up, volunteers walked at an intensity sufficient to maintain the HR in corresponding values to VT intensity for the next 40 min, and in the last 10 min they performed cooldown exercises. All participants used an HR monitor (Suunto, Ambit, Finland) for monitoring HR during all training sessions, and sessions were supervised at all times by a coach to ensure the HR had been maintained at the target value. The sessions were performed at the Olympic Training and Research Center (Centro Olímpico de Treinamento e Pesquisa; COTP, São Paulo, Brazil), in groups (five individuals per group).

**Table 1** List of *Pilates* exercises and apparatus.

| Name of exercises | Number of repetitions | Evolution in weeks |
|---|---|---|
| **Mat Pilates Program[a]** | | |
| Hundred | 1 | 4th–8th |
| Single leg circles, single leg stretch, double leg stretch, spine Stretch, saw, side kick | 3–5 | 1st–8th |
| Single leg kick, double leg kick, spine twist | 4–6 | 2nd–8th |
| Single straight leg stretch, criss-cross | 5–10 | 4th–8th |
| Shoulder bridge, leg pull front | 3–6 | 4th–8th |
| **Program on Reformer[a]** | | |
| Footwork, tendon stretch, pelvic lift, running, | 5–10 | 1st–8th |
| Short box series: round, flat back, twist | 3–5 | 4th–8th |
| Long box series: pulling straps, T | 5–8 | 4th–8th |
| Knee stretch series: round, arched back | 5–8 | 4th–8th |
| Long stretch series: front, elephant, down stretch | 5–8 | 4th–8th |
| Chest expansion kneeling, reverse chest expansion, splits side, stomach massage, flat back, reach, twist | 5–8 | 4th–8th |
| **Program on Cadillac[a]** | | |
| Roll down bar, breathing, chest expansion, thigh stretch | 3–5 | 1st–8th |
| Push through seated front, spread eagle | 3–5 | 1st–8th |
| Arm spring series (supine), leg spring series (supine) | 3–5 | 1st–8th |
| Side lying series | 5–10 | 4th–8th |
| Standing on floor: punching, salute, hug-a-tree, butterfly | 3–6 | 4th–8th |
| **Program on Chair[a]** | | |
| Double leg pumps, single leg pumps, standing leg pump front, side, crossover | 5–10 | 1st–8th |
| Washer woman, swan front, seated mermaid | 3–5 | 4th–8th |
| Frog lying flat, single leg pump lying flat, Achilles stretch | 5–10 | 1st–8th |
| **Program on Ladder Barrel[b]** | | |
| Horse back | 3 | 4th–8th |
| Short box series: round, flat back and twist | 3–5 | 4th–8th |
| **Program with Dumbbells–1kg[b]** | | |
| Wall: circles, sliding, rolling down | 3–5 | 1st–8th |
| Zip-up, chest expansion, shaving the head, arm circles, biceps curls (I, II), triceps extension, the bug, boxing, lunges | 5–10 | 2nd–8th |
| **Program with Magic Circle[a]** | | |
| Arm work, leg work and head work series | 5–10 | 1st–8th |

**Notes.**

[a] List of *Pilates* Exercises & Equipment. *Lessen (2014)*.
[b] *Siler & Turlington (2000)*. Details of the study protocol can be found on http://dx.doi.org/10.17504/protocols.io.p5wdq7e.

## Statistical analyses

Statistical analyses were performed using the Statistica software (Statsoft, Inc., version 6.0 for Windows, Tulsa, OK, USA). Data were expressed as mean $\pm$ standard deviation (SD). Variable distribution was tested by the Kolmogorov–Smirnov test, and variability by the Levene test. The sample size was planned to identify a difference between training groups

of at least 2% of fat mass in a relatively homogeneous group of healthy adults. To this end, we used data from a pilot study. The target sample size was 20 participants in each group (alpha 0.05 and power 0.80). After data collection, a two-way ANOVA for time vs. group of fat mass (%) and the power observed (alpha = 0.05) yielded the interaction effect of 0.87; therefore, we conclude that the sample size was sufficient for subsequent analysis.

To verify the effects of the intervention programs, a two-way ANCOVA, conducted by using age as a covariate, was used to assess the groups (control vs. aerobic vs. *Pilates*) and the time (pre- vs. post-training) differences for the following measures: anthropometric and body composition; waist and hip circumferences; abdominal, trunk, chair, stair and flexibility tests; and isokinetic and cardiorespiratory measurements. Newman–Keuls post hoc test procedures were used to identify specific differences. In the absence of interactions, only main effects were analyzed. Where age presented no significant difference for the variable analyzed in ANCOVA test, a two-way ANOVA was used.

In order to compare the HR and RPE reached during the sessions between the two intervention groups we used an unpaired Student's *t* test. Statistical significance was set at an $\alpha$ of 0.05 for all statistical procedures.

Calculation of effect size ($d$) was adopted in addition to the traditional statistical approaches. The measures of the effect size for changes in outcome were calculated by dividing the mean difference by the SD of the pre-training measurement. By calculating effect sizes, the magnitude of any change was judged according to the following criteria: $d = 0.2$ was considered a "small" effect size; $d = 0.5$ represented a "medium" effect size; and $d = 0.8$ was a "large" effect size (*Cohen, 1988*). Percent changes and 90% confidence intervals were also calculated.

## RESULTS

All volunteers performed 24 exercise sessions. The two-way ANOVA (group vs. time) revealed that there were no significant differences (pre- vs. post-training) in calorie intake for all groups [$F(2, 57) = 0.02744, p = 0.97$]. The target HR training for the aerobic group was $123 \pm 11$ bpm, which represents about $78.0 \pm 7.3\%$ of the maximal HR ($160 \pm 16$ bpm) reached during the cardiorespiratory maximal treadmill test. The *Pilates* group reached an HR training of $86 \pm 6$ bpm, which represents about $62.7 \pm 13.2\%$ of the maximal HR reached in the cardiopulmonary treadmill test. Thus, HR during training was significantly higher ($\Delta\% = +42.2\%$) in the aerobic group than in the *Pilates* group ($p < 0.0001$; see Table 2). Moreover, the internal training load was also higher ($\Delta\% = +31.3\%$) in the aerobic group than in the *Pilates* group ($p = 0.04$; see Table 2).

Two-way ANCOVA (group vs. time), using age as a covariate, revealed that the age presented a significant effect for abdominal strength test, flexibility test, stair test, peak torque for dominant and non-dominant extensor muscles at 240°/s and HR assessed at VT, RCP and maximal exercise. Therefore, the results presented for these variables were from ANCOVA using age as a covariate. Results presented for the other variables (BMI, total body mass, fat mass, fat free mass, waist and hip circumference, trunk endurance test, chair test, peak torque for extensor and flexors muscles at 60°/s and 240°/s and for flexors

**Table 2  Average heart rate and internal training load during training for *Pilates* and aerobic groups.**

| Variables | *Pilates* (n = 22) | Aerobic (n = 21) | Δ% | p value | d value | 90% CI |
|---|---|---|---|---|---|---|
| Average HR (bpm) | 86 ± 6 | 123 ± 11[*] | +42.2 | <0.0001 | 4.1 | 2.5–5.7 |
| Internal training load | 192 ± 69 | 252 ± 106[*] | +31.3 | 0.04 | 0.67 | 0.14–1.2 |

Notes.
  Data are mean ± standard deviation.
  HR, heart rate; bpm, beats per minute; d, Cohen d effect size; CI, confidence interval.
  [a] Aerobic > *Pilates*.

at 240/s and $\dot{V}O_2$ at VT, RCP, or maximal exercise) were from ANOVA (group vs. time), once age presented no significant difference in the analysis of these variables.

There were no significant differences in body composition, anthropometric measurements, and functional tasks among the three groups at baseline (Table 3), except for the abdominal endurance test [$F(2, 57) = 3.69, p = 0.031$]. The aerobic group had higher values than both the control group ($p = 0.02$) and the *Pilates* group ($p = 0.01$; see Table 3).

As expected, after an 8-week period, the control group showed no differences in any of the outcome variables., except for abdominal test performance with higher values after the experimental period ($p = 0.02$, $d = 0.57$ and $\Delta\% = +9.3 \pm 14.8\%$). Total body mass did not change in any of the groups and despite the significant improvement, in fat mass ($\Delta\% = -3.7 \pm 5.2\%$) and lean mass ($\Delta\% = +3.2 \pm 5.0\%$) in the *Pilates* group, its effect was classified as small ($d = 0.19$). There was no difference in body composition between groups after the intervention period ($p > 0.05$). As expected, the *Pilates* and aerobic groups showed significant improvements in waist circumference ($p = 0.0009, p = 0.0001$, respectively), hip circumference ($p = 0.0001, p = 0.001$, respectively), abdominal test ($p = 0.0001, p = 0.003$, respectively), and chair test ($p = 0.0001, p = 0.005$, respectively) and stair test ($p = 0.003, p = 0.046$, respectively). Trunk test and flexibility test improved only in the *Pilates* group ($p = 0.0003$ and $p = 0.0001$, respectively; see Table 3).

As both the *Pilates* and aerobic groups showed significant improvements in some outcome variables after the intervention, the effect size was also measured and the difference between pre- and post-tests was also expressed as perceptual difference (Table 3). The effect size and perceptual difference of the *Pilates* and aerobic training in waist ($d = 0.27$, $\Delta\% = -2.6 \pm 2.9\%$ and $d = 0.37$, $\Delta\% = -3.3 \pm 3.7\%$, respectively) and hip circumference ($d = 0.17$, $\Delta\% = -1.5 \pm 1.6\%$ and $d = 0.14$, $\Delta\% = -0.8 \pm 1.0\%$, respectively) were classified as medium for waist values and small for hip values. There was also no significant difference between groups after intervention ($p > 0.05$). Despite a significant improvement in the abdominal test in all three groups ($d = 1.69$, $\Delta\% = 25.6 \pm 13.5$; $d = 0.95$, $\Delta\% = 11.0 \pm 16.7\%$ and $d = 0.58$, $\Delta\% = 9.3 \pm 14.8\%$ for *Pilates*, aerobic and control groups, respectively), the effect size and perceptual difference for the *Pilates* group were greater than the effect size and perceptual difference for the aerobic group, which also were greater than these variables for the control group. Moreover, after the intervention period, the control group showed significantly lower values for the abdominal test than the *Pilates* ($p = 0.005$) and aerobic groups ($p = 0.004$).

**Table 3  Effects of 8 weeks of *Pilates* or aerobic training on variables assessed.**

| | Pre | Post | Δ% | p value | d value | 90% CI |
|---|---|---|---|---|---|---|
| **Pilates group (n = 22)** | | | | | | |
| BMI (kg/m²) | 31.9 ± 3.9 | 31.6 ± 4.0[a] | −1.1 ± 1.8 | 0.009 | 0.07 | 0.01–0.13 |
| Total body mass (kg) | 81.0 ± 10.0 | 80.8 ± 10.2 | −0.5 ± 1.4 | 0.19 | 0.01 | −0.01–0.03 |
| Fat mass (%) | 50.9 ± 6.1[b] | 49.1 ± 5.8[a] | −3.7 ± 5.2 | 0.0001 | 0.19 | 0.11–0.27 |
| Lean mass (kg) | 38.4 ± 6.5 | 39.7 ± 6.6[a] | 3.2 ± 5.0 | 0.0005 | 0.19 | 0.11–0.27 |
| Waist circumference (cm) | 105.6 ± 9.8 | 102.9 ± 9.9[a] | −2.6 ± 2.9 | 0.0009 | 0.27 | 0.14–0.40 |
| Hip circumference (cm) | 115.7 ± 9.4 | 114.0 ± 10.1[a] | −1.5 ± 1.6 | 0.0001 | 0.17 | 0.1–0.24 |
| Abdominal test (no. of repetitions) | 30.7 ± 6.3 | 41.6 ± 6.6[a] | 25.6 ± 13.5 | 0.0001 | 1.69 | 1.0–2.40 |
| Trunk extensor test (s) | 89.5 ± 42.8 | 129.4 ± 55.7[a] | 28.7 ± 23.6 | 0.0003 | 0.80 | 0.46–1.10 |
| Chair test (s) | 24.5 ± 3.3 | 19.6 ± 3.3[a] | −26.9 ± 27.9 | 0.0001 | 1.48 | 0.89–2.10 |
| Stair test (s) | 25.3 ± 3.4 | 22.8 ± 3.0[a] | −9.8 ± 7.9 | 0.003 | 0.78 | 0.40–1.20 |
| Flexibility (cm) | 21.5 ± 10.7 | 25.6 ± 9.4[a] | 21.8 ± 21.6 | 0.0001 | 0.41 | 0.25–0.57 |
| **Aerobic group (n = 21)** | | | | | | |
| BMI (kg/m²) | 30.3 ± 3.3 | 30.1 ± 3.4 | −0.6 ± 1.8 | 0.12 | 0.05 | −0.01–0.10 |
| Total body mass (kg) | 77.7 ± 10.6 | 77.4 ± 10.6 | −0.5 ± 1.8 | 0.20 | 0.03 | −0.01–0.07 |
| Fat mass (%) | 46.4 ± 6.2 | 45.9 ± 6.4 | −1.2 ± 2.4 | 0.60 | 0.05 | −0.03–0.14 |
| Lean mass (kg) | 40.0 ± 5.5 | 40.2 ± 5.6 | 0.5 ± 2.7 | 0.49 | 0.03 | −0.04–0.10 |
| Waist circumference (cm) | 102.7 ± 8.4 | 99.5 ± 8.8[a] | −3.3 ± 3.7 | 0.0001 | 0.37 | 0.22–0.52 |
| Hip circumference (cm) | 111.6 ± 7.2 | 110.6 ± 7.3[a] | −0.8 ± 1.0 | 0.001 | 0.14 | 0.07–0.21 |
| Abdominal test (no. of repetitions) | 36.9 ± 5.6[c] | 42.1 ± 5.3[a] | 11.0 ± 16.7 | 0.003 | 0.95 | 0.46–1.40 |
| Trunk extensor test (s) | 102.6 ± 39.0 | 117.2 ± 53.3 | 3.4 ± 40.8 | 0.09 | 0.31 | 0.02–0.60 |
| Chair test (s) | 22.7 ± 3.4 | 20.5 ± 2.3[a] | −11.3 ± 16.5 | 0.005 | 0.75 | 0.28–1.20 |
| Stair test (s) | 24.3 ± 3.1 | 22.8 ± 4.1[a] | −7.4 ± 10.5 | 0.046 | 0.41 | 0.10–0.72 |
| Flexibility (cm) | 20.6 ± 9.9 | 21.7 ± 9.5 | 10.4 ± 23.9 | 0.10 | 0.11 | 0.01–0.21 |
| **Control group (n = 17)** | | | | | | |
| BMI (kg/m²) | 31.0 ± 4.1 | 31.1 ± 3.9 | 0.4 ± 1.1 | 0.27 | 0.02 | −0.01–0.05 |
| Total body mass (kg) | 82.6 ± 16.6 | 82.9 ± 16.4 | 0.4 ± 1.1 | 0.34 | 0.01 | −0.01–0.02 |
| Fat mass (%) | 45.7 ± 7.7 | 46.0 ± 7.6 | 0.6 ± 3.9 | 0.49 | 0.04 | −0.01–0.09 |
| Lean mass (kg) | 43.6 ± 10.5 | 43.6 ± 10.2 | 0.0 ± 3.1 | 0.93 | 0 | 0 |
| Waist circumference (cm) | 102.9 ± 10.5 | 103.5 ± 10.8 | 0.5 ± 1.4 | 0.41 | 0.02 | −0.02–0.06 |
| Hip circumference (cm) | 109.2 ± 9.8 | 109.4 ± 10.0 | 0.3 ± 0.5 | 0.26 | 0.02 | −0.01–0.05 |
| Abdominal test (no. of repetitions) | 31.9 ± 6.1 | 35.6 ± 6.7[d, a] | 9.3 ± 14.8 | 0.02 | 0.57 | 0.17–0.97 |
| Trunk extensor test (s) | 90.2 ± 25.6 | 91.4 ± 42.6 | −9.1 ± 40.0 | 0.92 | 0.03 | −0.47–0.53 |
| Chair test (s) | 23.1 ± 4.3 | 22.9 ± 3.5[d] | −1.3 ± 14.8 | 0.78 | 0.05 | −0.25–0.35 |
| Stair test (s) | 23.7 ± 3.5 | 23.9 ± 3.3 | 0.5 ± 11.4 | 0.65 | 0.05 | −0.13–0.23 |
| Flexibility (cm) | 19.0 ± 12.0 | 19.0 ± 11.8 | 2.1 ± 13.2 | 1.00 | 0 | 0 |

**Notes.**
Data are mean ± standard deviation.

BMI, body mass index; $d$, Cohen $d$ effect size; CI, confidence interval.

[a] $p < 0.05$ (different from the pre-training for the same group).

[b] $p < 0.05$ (pre-training values for Control group ≠ pre-training values for *Pilates* group).

[c] $p < 0.05$ (pre-training values for Aerobic group ≠ pre-training values for *Pilates* and Control groups).

[d] $p < 0.05$ (pos-training values for Control group ≠ pos-training values for *Pilates* and Aerobic groups).

[d] $p < 0.05$ (pos-training values for Control group ≠ pos-training values for *Pilates* group).

Results of the two evaluated functional tasks differed across the groups. On the chair test $[F(2, 57) = 8.49, p = 0.00]$, the improvement observed in the *Pilates* group ($\Delta\% = -26.9 \pm 27.9\%$, $d = 1.48$, p=0.0001) was greater than that seen in the aerobic group ($\Delta\% = -11.3 \pm 16.5\%$, $p = 0.005$, $d = 0.75$). Moreover, only the *Pilates* group showed a significant improvement performance in the chair test than the control group ($p = 0.02$, $\Delta\% = 30.0 \pm 27.9\%$) after the intervention period. In the same direction, the stair test $[F(2, 57) = 4.00, p = 0.024]$. The improvement observed in the *Pilates* group was also greater than that seen in the aerobic group ($d = 0.78$, $\Delta\% = 9.8 \pm 7.9\%$ and $d = 0.41$, $\Delta\% = 7.4 \pm 10.5\%$, respectively). However, there were no significant differences between groups after the intervention period.

Isokinetic PT for dominant ($p = 0.02$) and non-dominant ($p = 0.02$) extensor muscles ($60°/s$) were significantly lower for the *Pilates* group than for controls at baseline. However, after the 8-week period, neither isokinetic PT of flexor nor for extensor muscles in both dominant and non-dominant limbs presented significant differences (pre- vs. post-intervention) in any of the three groups at any angular speed tested ($60°/s$ and $240°/s$; see Table 4).

Regarding cardiorespiratory fitness prior to the intervention period at baseline, there were no significant differences between the three groups in $\dot{V}O_2$ values at VT, RCP, or max values (p >0.05). The *Pilates* group presented lower HR values than the aerobic ($p = 0.0002$) and control groups ($p = 0.0002$) in VT, and than the aerobic ($p = 0.004$) and control (p=0.005) groups in RCP, and lower values than the aerobic ($p = 0.03$) and control ($p = 0.001$) group in maximal effort. After the training period, the *Pilates* group presented a significant increase in $\dot{V}O_2$ and in HR reached at VT ($p = 0.001$ and $p = 0.0005$, respectively), RCP ($p = 0.01$ and $p = 0.01$, respectively) and at maximum effort ($p = 0.01$ and $p = 0.003$, respectively) (Table 5). The control group presented significant different HR in maximal effort after and before the intervention period ($p = 0.02$). The aerobic group presented no significant difference in any cardiorespiratory variables measured (Table 5). Despite the significant increase in $\dot{V}O_2$ at VT ($p = 0.001$, $d = 0.60$, $\Delta\% = 11.6 \pm 17.6\%$), RCP ($p = 0.01$, $d = 0.48$, $\Delta\% = 7.9 \pm 19.5\%$) and maximum values ($p = 0.01$, $d = 0.33$, $\Delta\% = 19.9 \pm 15.5\%$) observed only in the *Pilates* group, after the experimental period there were no differences between the groups.

## DISCUSSION

This study compared the impact of *Pilates* and aerobic training on cardiorespiratory fitness, isokinetic muscular strength, body composition, flexibility, trunk and abdominal endurance, and functional task outcomes for overweight/obese individuals. Our main findings were that BMI, lean mass, fat mass, waist circumference, hip circumference, abdominal and trunk strength tests, chair and stair test, flexibility and $\dot{V}O_2$ assessed at VT, RCP and at maximum effort improved in Pilates group. Conversely, only waist and hip circumference, abdominal strength test, and chair and stair test improved in aerobic training group. Abdominal strength test was lower in control group than in both intervention groups in post-tests and chair test were lower in control than in Pilates group in post-tests.

**Table 4  Isokinetic strength measured at 60°/s and at 240°/s, pre- and post-training, for *Pilates*, aerobic, and control groups.**

| Variables | Pre | Post | Δ% | p value | d values | 90% CI |
|---|---|---|---|---|---|---|
| *Pilates* (*n* = 22) | | | | | | |
| 60°/s | | | | | | |
| PT/BM extensor dominant (%) | 128.9 ± 37.6[b] | 138.6 ± 36.2 | 5.5 ± 12.2 | 0.34 | 0.26 | −0.19–0.71 |
| PT/BM extensor non-dominant (%) | 124.4 ± 36.2 | 119.2 ± 39.3 | −0.9 ± 15.5 | 0.83 | 0.13 | −0.88–1.10 |
| PT/BM flexor dominant (%) | 65.7 ± 27.4 | 72.6 ± 24.7 | 5.0 ± 28.6 | 1.00 | 0.26 | −34.0–35.0 |
| PT/BM flexor non-dominant (%) | 62.6 ± 19.3 | 67.4 ± 16.5 | 5.1 ± 17.9 | 0.98 | 0.25 | −16.0–17.0 |
| 240°/s | | | | | | |
| PT/BM extensor dominant (%) | 75.8 ± 21.9[b] | 76.2 ± 23.7 | 6.6 ± 13.1 | 0.83 | 0.02 | −0.14–0.18 |
| PT/BM extensor non-dominant (%) | 72.4 ± 21.5[b] | 74.3 ± 18.1 | 2.1 ± 19.3 | 0.58 | 0.01 | −0.02–0.04 |
| PT/BM flexor dominant (%) | 65.8 ± 27.4 | 51.1 ± 14.0 | 8.3 ± 22.2 | 0.50 | 0.67 | −0.98–2.30 |
| PT/BM flexor non-dominant (%) | 62.6 ± 19.3 | 46.3 ± 12.4 | 7.9 ± 19.1 | 0.29 | 1.00 | −0.57–2.60 |
| Aerobic (*n* = 21) | | | | | | |
| 60°/s | | | | | | |
| PT/BM extensor dominant (%) | 146.5 ± 25.0 | 146.4 ± 24.2 | −1.1 ± 15.9 | 1.00 | 0.00 | 0 |
| PT/BM extensor non-dominant (%) | 137.5 ± 38.1 | 135.9 ± 36.2 | −2.2 ± 16.5 | 0.99 | 0.04 | −5.30–5.40 |
| PT/BM flexor dominant (%) | 75.5 ± 28.9 | 72.6 ± 19.1 | −5.7 ± 27.4 | 0.99 | 0.12 | −16.0–16.0 |
| PT/BM flexor non-dominant (%) | 63.1 ± 19.1 | 68.9 ± 17.8 | 8.2 ± 16.1 | 0.99 | 0.31 | −41.0–41.0 |
| 240°/s | | | | | | |
| PT/BM extensor dominant (%) | 90.3 ± 14.7 | 89.2 ± 16.9 | −2.2 ± 10.9 | 0.98 | 0.07 | −4.60–4.70 |
| PT/BM extensor non-dominant (%) | 87.0 ± 18.5 | 85.3 ± 20.6 | −4.5 ± 20.9 | 0.17 | 0.08 | −0.02–0.18 |
| PT/BM flexor dominant (%) | 52.8 ± 9.5 | 55.7 ± 14.4 | 2.4 ± 19.2 | 0.84 | 0.23 | −1.70–2.10 |
| PT/BM flexor non-dominant (%) | 52.5 ± 10.0 | 50.9 ± 10.9 | −5.5 ± 19.3 | 0.95 | 0.15 | −3.80–4.10 |
| Control (*n* = 17) | | | | | | |
| 60°/s | | | | | | |
| PT/BM extensor dominant (%) | 165.9 ± 36.6 | 163.6 ± 38.3 | −3.0 ± 112 | 0.99 | 0.06 | −7.90–8.00 |
| PT/BM extensor non-dominant (%) | 157.1 ± 34.8 | 161.2 ± 38.9 | 0.3 ± 13.7 | 0.98 | 0.11 | −7.20–7.40 |
| PT/BM flexor dominant (%) | 76.9 ± 18.5 | 80.4 ± 20.8 | 2.4 ± 11.4 | 0.89 | 0.18 | −2.00–2.30 |
| PT/BM flexor non-dominant (%) | 74.2 ± 21.6 | 74.4 ± 22.5 | −0.9 ± 16.3 | 0.32 | 0.01 | −0.01–0.02 |
| 240°/s | | | | | | |
| PT/BM extensor dominant (%) | 99.0 ± 19.9 | 103.5 ± 21.4 | 3.7 ± 6.9 | 0.98 | 0.22 | −14.0–15.0 |
| PT/BM extensor non-dominant (%) | 95.6 ± 18.9 | 102.4 ± 22.9 | 4.9 ± 10.4 | 1.00 | 0.32 | −42.0–43.0 |
| PT/BM flexor dominant (%) | 53.7 ± 15.5 | 55.5 ± 11.4 | 2.9 ± 12.2 | 0.98 | 0.13 | −8.50–8.80 |
| PT/BM flexor non-dominant (%) | 53.6 ± 12.6 | 53.3 ± 13.4 | −3.0 ± 20.6 | 0.99 | 0.02 | −2.60–2.70 |

**Notes.**
Data are mean ± standard deviation.
PT, peak torque; BM, body mass; *d*, Cohen *d* effect size; CI, confidence interval.
*$p < 0.05$ (pre vs. post).
#$p < 0.05$ (*Pilates* vs. Control).

## Body composition effects

Traditionally, body mass reduction by decreasing fat mass is the main goal in obesity treatment, and several studies have investigated the effects of exercise for that purpose, although the importance of physical activity for weight loss assessed by body weight or BMI is controversial (*Pedersen & Saltin, 2015*). The aerobic exercise protocol employed in

**Table 5  Effects of 8 weeks of *Pilates* or aerobic training on cardiorespiratory maximal treadmill test results for *Pilates*, Aerobic and Control groups.**

| Variables | Pre | Post | Δ% | *p* value | *d* value | 90% CI |
|---|---|---|---|---|---|---|
| ***Pilates*** (*n* = 22) | | | | | | |
| $\dot{V}O_2$ in VT (mL/kg/min) | 15.0 ± 2.6 | 17.2 ± 3.4[*] | 11.6 ± 17.6 | <0.01 | 0.60 | 0.31–0.89 |
| HR in VT (bpm) | 109 ± 13[b] | 122 ± 15[*] | 10 ± 11 | <0.01 | 0.90 | 0.47–1.30 |
| $\dot{V}O_2$ in RCP (mL/kg/min) | 19.4 ± 3.6 | 21.1 ± 4.1[*] | 7.9 ± 19.5 | 0.01 | 0.48 | 0.18–0.78 |
| HR in RCP (bpm) | 130 ± 15[b] | 139 ± 15 | 6.0 ± 9.6 | 0.01 | 0.56 | 0.21–0.91 |
| $\dot{V}O_2$max (mL/kg/min) | 22.4 ± 4.9 | 24.2 ± 4.5[*] | 19.9 ± 15.5 | 0.01 | 0.33 | 0.12–0.54 |
| HRmax (bpm) | 146 ± 21[b] | 155 ± 16[*] | 6.2 ± 11.7 | <0.01 | 0.48 | 0.24–0.72 |
| **Aerobic** (*n* = 21) | | | | | | |
| $\dot{V}O_2$ in VT (mL/kg/min) | 18.7 ± 2.7 | 20.1 ± 3.2 | 6.1 ± 10.2 | 0.22 | 0.38 | 0.05–0.71 |
| HR in VT (bpm) | 126 ± 10 | 127 ± 17 | −0.7 ± 16.6 | 0.93 | 0.09 | −0.16–1.8 |
| $\dot{V}O_2$ in RCP (mL/kg/min) | 23.2 ± 3.8 | 24.3 ± 3.8 | 3.0 ± 12.3 | 0.72 | 0.21 | −0.13–0.55 |
| HR in RCP (bpm) | 146 ± 16 | 145 ± 19 | −1.8 ± 11.0 | 0.91 | 0.03 | −0.09–0.15 |
| $\dot{V}O_2$max (mL/kg/min) | 26.8 ± 4.3 | 27.0 ± 4.4 | 14.8 ± 10.5 | 0.64 | 0.13 | −0.11–0.37 |
| HRmax (bpm) | 160 ± 16 | 158 ± 19 | −1.7 ± 6.6 | 0.48 | 0.12 | −0.16–0.40 |
| **Control** (*n* = 17) | | | | | | |
| $\dot{V}O_2$ in VT (mL/kg/min) | 17.9 ± 3.9 | 18.7 ± 4.3 | 3.4 ± 14.7 | 0.18 | 0.19 | −0.04–0.42 |
| HR in VT (bpm) | 127 ± 11 | 128 ± 10 | 0.4 ± 10.0 | 0.94 | 0.10 | −2.10–2.30 |
| $\dot{V}O_2$ in RCP (mL/kg/min) | 22.3 ± 4.9 | 22.4 ± 4.5 | −1.2 ± 10.1 | 0.73 | 0.04 | −0.15–0.23 |
| HR in RCP (bpm) | 146 ± 13 | 145 ± 10 | −0.8 ± 6.3 | 0.92 | 0.05 | −0.78–0.88 |
| $\dot{V}O_2$max(mL/kg/min) | 25.9 ± 5.4 | 25.2 ± 4.7 | 9.8 ± 11.5 | 0.31 | 0.08 | −0.05–0.21 |
| HRmax (bpm) | 165 ± 12 | 156 ± 12[*] | −6.2 ± 6.5 | 0.02 | 0.74 | 0.22–1,30 |

Notes.

Data are mean ± standard deviation.

$\dot{V}O_2$, oxygen uptake; $\dot{V}O_2$ max, maximal oxygen uptake; HR, heart rate; HRmax, maximal heart rate; VT, ventilatory threshold; RCP, respiratory compensation point; *d*, Cohen *d* effect size; CI, confidence interval.

[a] *p* < 0.05 (pre vs. post).
[b] *p* < 0.05 (*Pilates* < Aerobic and Control).

the present study promoted no difference in body composition. *Cheema et al. (2015)* also failed to find a significant loss of fat mass percentage after an aerobic program performed at moderate intensity for 12 weeks. Most likely, the amount of exercise (volume and/or intensity) necessary to produce weight loss needed to be greater than the one implemented in our study, and a restrictive diet would also be necessary for body mass loss.

Changes induced by the *Pilates* were greater than those promoted by aerobic training, as the *Pilates* group presented a significant increase in lean mass and decrease in fat mass. These results suggest that *Pilates* may be an interesting alternative to traditional aerobic training in order to improve body composition (*Jago et al., 2006*; *Rogers & Gibson, 2009*; *Şavkin & Aslan, 2017*). Conversely, due to the small effect size, the results should be viewed with caution. Moreover, there were no differences between groups after the intervention period. Previous studies also show effective results for *Pilates* in reducing fat mass and increasing fat free mass (*Rogers & Gibson, 2009*; *Şavkin & Aslan, 2017*). Therefore, despite the small effect size observed in our study after the *Pilates*, our results are in line with the literature. Conversely, these results are not a consensus. *Segal, Hein & Basford (2004)* showed no

significant difference in body composition after *Pilates*. Methodological differences may be responsible for these conflicting results; different indirect methods—which tend to be inaccurate—used to assess body composition (*Rogers & Gibson, 2009*); a lack of information about the instructors' certification (*Segal, Hein & Basford, 2004*; *Rogers & Gibson, 2009*); and significant differences in the protocol used (*Pilates* mats or apparatus, sessions per week and program duration); associated with a lack of diet control (*Segal, Hein & Basford, 2004*) hamper definitive conclusions on this topic. Considering that DXA, which has been previously demonstrated to be reliable (*Choi et al., 2015*) was used in our study to measure body composition, it can be concluded that *Pilates* may be considered an alternative exercise program for improving body composition.

### Functional tasks tests

Both the aerobic and *Pilates* groups showed significant improvement on both tests, unlike the control group. However, the improvements on the chair and stair tests in the *Pilates* group were greater (i.e., had a higher effect size) than in the aerobic group. Moreover, only the *Pilates* group presented significantly better results in the chair test than the control group after the intervention. This improvement in functional tasks presented in both experimental groups is of fundamental importance for independent living and mobility (*Miller & Wolfe, 2008*).

### Abdominal and trunk strength test and flexibility test

The abdominal and trunk strength tests performed in this study have been found to be reliable field tests (*Sparling, Millard-Stafford & Snow, 1997*; *Demoulin et al., 2006*). Moreover, sit and reach test has also been found to be reliable (*Bozic et al., 2010*). All three groups had higher values for the abdominal test after the 8-week intervention, suggesting that there was a learning effect on this test demonstrated by the improvement in the control group. Notwithstanding, both the intervention groups presented significantly higher results in the abdominal test after the intervention period than the control group. Demonstrating that both the aerobic and *Pilates* interventions were effective to improve abdominal strength. It is important to notice that the *Pilates* was more effective to improve abdominal muscle strength, demonstrated by the higher effect size value. In the same direction, trunk extensor muscle endurance showed a significant improvement only in the *Pilates* group, in the same way as flexibility. In general, these results were expected, since the *Pilates* method focuses on these muscles through posture and core exercises (*Şavkin & Aslan, 2017*). Although the effect size of this improvement was classified as large, after the intervention period there were no significant differences between groups. The individuals from each group were quite different and the SD of the measurements was very high; this may be the reason for the absence of significant difference between groups after the intervention, despite the improvement observed only in the *Pilates* group. We therefore conclude that for functional tasks (chair and stair tests), trunk and abdominal endurance tests, both training programs should be considered, but the *Pilates* can produce better results.

### Isokinetic strength test

Isokinetic dynamometry is a method commonly used to assess muscle performance, both in research and in clinical practice (*Granata, Abel & Damiano, 2000*; *Fleury et al., 2011*; *Andrade et al., 2013*). As has been previously demonstrated, isokinetic dynamometry is a reliable and sensitive instrument to evaluate therapeutic intervention outcomes, including in young or elderly people (*Capranica et al., 1998*; *Eitzen, Hakestad & Risberg, 2012*; *Andrade et al., 2013*).

Isokinetic muscular strength, mainly in the *Pilates* group, was expected to show a significant improvement, but there were no changes in isokinetic PT for either muscular group assessed (knee flexor and extensor muscles) at 60°/s and 240°/s. Since *Pilates* exercises are not isokinetic exercises, the mode of contraction and the angular speed are different, and this lack of specificity of the evaluation (isokinetic dynamometer) may contribute to unexpected results. Another possible explanation for the absence of difference in isokinetic strength results after the intervention in the *Pilates* group may be related to the lower power of isokinetic analysis. For flexor and extensor PT values, the observed power ranged from 0.06 to 0.43. Lower power means that there is a high probability of concluding there is no effect when one actually exists. In other words, the number of subjects in this study might be not sufficient to ensure (to a particular degree of certainty) that the interventions produced no effects on muscular isokinetic strength.

### Cardiorespiratory maximal treadmill test

In the present study, the *Pilates* group showed a significant increase in $\dot{V}O_2$max after the intervention, which did not occur in the aerobic group, although the *Pilates* intervention was less intense than the walking intervention (internal training load 192 ± 69 vs. 252 ± 106). The lack of $\dot{V}O_2$max improvement after aerobic training was in accordance with previous data (*Shinkai et al., 1994*; *Fiorilli et al., 2017*). *Shinkai et al. (1994)* studied the effects on $\dot{V}O_2$max of a 12-week aerobic program plus voluntary food restriction for mildly obese middle-age women. The authors observed no change in absolute $\dot{V}O_2$max/peak values (L/min), and a significant increase only in relative $\dot{V}O_2$max/peak values (mL/kg/min) after the program, which could be attributed to the significant body mass loss observed, instead of a real $\dot{V}O_2$max improvement. *Fiorilli et al. (2017)* also studied the effects on $\dot{V}O_2$max of an aerobic or resistance program. Interestingly, their results showed higher $\dot{V}O_2$max values for the resistance-training group than for the aerobic training group.

With regards to $\dot{V}O_2$max, this variable can be derived from the Fick equation: $\dot{V}O_2\max = COmaxx(Ca - C\bar{v})O_2\max$. Therefore, $\dot{V}O_2$max depends on the maximal CO (cardiac output) and the maximal $(Ca - CO\bar{v}_2)O_2max)$ (difference between oxygen concentration taken from arterial and mixed venous blood). In most situations, $(Ca - C\bar{v}_2)O_2max)$ does not vary much between individuals, so the $\dot{V}O_2$ is usually limited by the maximal CO (*Bassett & Howley, 2000*); however, in some special situations, such as detraining, very sedentary people or muscular dystrophy, it may be limited by a low maximal $(Ca-C\bar{v}_2)O_2$, resulting from a low muscular mass capacity.

It is probable that the $\dot{V}O_2$max/peak improved in the *Pilates* group as a consequence of a higher maximal CO resulting from the higher maximal HR reached at the end of the

cardiopulmonary test. Moreover, as indicated above, in detraining or very sedentary people, the $\dot{V}O_2$max may be limited by a low maximal (Ca-CO $\bar{v}_2$) $O_2$. Considering that the *Pilates* group exercises are focused on muscular conditioning, it is possible that the increase in the $\dot{V}O_2$max/peak resulted from the observed muscular mass improvement after the program, too. However, it was not possible to state that the aerobic capacity of the muscular mass has increased. The aerobic group failed to show an improvement in the $\dot{V}O_2$max/peak after the training period. During the walking session, the HR mean was $123 \pm 11$ bpm (HR corresponding to the VT), which represents about 78% of the maximal HR for this group, which is in accordance with aerobic program recommendations (training intensity greater than 60% of the HRmax, with 3–5 sessions/week for 30 min/session). Although moderate aerobic exercise has been extensively suggested for obese people (*Pedersen & Saltin, 2015*; *Burgess et al., 2017*; *Mabire et al., 2017*), our data were in line with other previous literature data (*Cheema et al., 2015*) which demonstrated that moderate walking performed for 3 months (4 times a week) was not effective to improve aerobic fitness in overweight and obese adults. It is possible that an even greater effort intensity or training period is necessary to produce an increase of maximal CO and/or maximal (Ca-CO $\bar{v}_2$) $O_2$ resulting in greater $\dot{V}O_2$max/peak. Other studies that used high-intensity interval training showed significant improvement in the $\dot{V}O_2$max/peak (*Cheema et al., 2015*; *Ruffino et al., 2017*), so exercise intensity may be crucial to improve the $\dot{V}O_2$max/peak.

Although there was no significant difference between groups after the intervention period, the improvement in $\dot{V}O_2$max/peak in the *Pilates* group was classified as medium, and considering the strong association between an increase in the $\dot{V}O_2$max/peak and reduced all-cause and cardiovascular disease mortality (*Lee et al., 2011*; *Barlow et al., 2012*), this is a meaningful benefit of *Pilates*.

VT is another aerobic index that has great practical application. Besides VT, prolonged exercise leads to progressive metabolic acidosis and exhaustion. VT depends on peripheral circulatory factors, specifically on the oxidative capacity of the contracting muscles (*Weltman, 1995*). In this study, the $\dot{V}O_2$ in the VT increased significantly only in the *Pilates* group ($d = 0.6$). Muscle conditioning is one of the objectives of the *Pilates*. Possibly better muscle conditioning, characterized by improved peripheral capillarization and muscle oxidative capacity, were responsible for the $\dot{V}O_2$ improvement measured at the VT. In order to better understand the $\dot{V}O_2$max/peak, VT, and RCP improvement in the *Pilates* group, future studies should evaluate muscular oxidative capacity. In addition, further studies, aiming to investigate the Pilates and walking training effects on more general health parameters, such as the risk of morbidity and mortality, or on muscular oxidative capacity should be developed.

## Strengths and limitations of this study

The external validity of the study was guaranteed through an extensive process of volunteer recruitment (social networks and within the medical school) and through the randomization process that divided the volunteers between the two intervention groups. One should, however, consider the cost of a *Pilates* practice program in a studio.

This aspect should be taken into consideration in the appointment of a physical health program for individuals who are overweight/obese.

A possible bias of this study is the inclusion of both men and women. However, an attenuating factor is that both intervention groups contained the same number of men.

Another possible bias is the lack of randomization between control and experimental groups. Twenty-five volunteers asked to participate in the control group, because the intervention protocol (*Pilates* or walking sessions) was during their working hours. If a random order were used, very intense encouragement would be necessary to guarantee adherence, creating an unreal situation. In order to avoid this bias, these volunteers were allocated to the control group. Choosing for non-randomization between the control and experimental groups could have created a bias as a result of the lack of homogeneity of the baseline data, which, fortunately, did not occur; there was no significant difference in baseline data for muscular performance (strength, endurance, and flexibility), body composition, or functional tasks outcomes.

The same intervention period of 8 weeks was performed for both Aerobic and Pilates groups. It was chosen as it was important that the two groups were submitted to the same training volume. Considering that the aerobic training is usually performed for 12 weeks in previous studies, perhaps other outcomes can be found with longer intervention periods.

## CONCLUSION

To our knowledge, this is the first study comparing *Pilates* with traditionally aerobic exercise for overweight or obese subjects. *Pilates* should be considered as an interesting alternative physical fitness program for individuals who are overweight or obese, since it can lead to significant improvements in body composition, general strength (trunk and abdominal), flexibility, ability to accomplish functional tasks, and cardiorespiratory fitness.

## ACKNOWLEDGEMENTS

We would like to thank all of the participants who volunteered their time to participate in the study, the Olympic Training and Research Center (Centro Olímpico de Treinamento e Pesquisa, COTP, São Paulo, Brazil), and the CGPA Pilates Studio (São Paulo, Brazil).

### Funding
This work was supported by the Fundação de Amparo à Pesquisa do Estado de São Paulo (FAPESP) (grant number: 2013/08245-2). Ricardo Borges Viana had a fellowship from the Coordenação de Aperfeiçoamento de Pessoal de Ensino Superior (CAPES/ - Coordination for the Improvement of Higher Education Personnel, Brazil). The funders had no role in study design, data collection and analysis, decision to publish, or preparation of the manuscript.

## Grant Disclosures

The following grant information was disclosed by the authors:

Fundação de Amparo à Pesquisa do Estado de São Paulo (FAPESP): 2013/08245-2.

Coordenação de Aperfeiçoamento de Pessoal de Ensino Superior (CAPES/ - Coordination for the Improvement of Higher Education Personnel, Brazil.

## Competing Interests

The authors declare there are no competing interests.

## Author Contributions

- Angeles Bonal Rosell Rayes conceived and designed the experiments, performed the experiments, analyzed the data, prepared figures and/or tables, authored or reviewed drafts of the paper, approved the final draft.
- Claudio Andre B. de Lira and Marilia S. Andrade conceived and designed the experiments, analyzed the data, contributed reagents/materials/analysis tools, prepared figures and/or tables, authored or reviewed drafts of the paper, approved the final draft.
- Ricardo B. Viana prepared figures and/or tables, authored or reviewed drafts of the paper, approved the final draft.
- Ana A. Benedito-Silva conceived and designed the experiments, analyzed the data, authored or reviewed drafts of the paper, approved the final draft.
- Rodrigo L. Vancini authored or reviewed drafts of the paper, approved the final draft.
- Naryana Mascarin performed the experiments, analyzed the data, authored or reviewed drafts of the paper, approved the final draft.

## Human Ethics

The following information was supplied relating to ethical approvals (i.e., approving body and any reference numbers):

The Federal University of São Paulo granted Ethical approval to carry out the study within its facilities (ethical Application Ref: 197.451).

## Clinical Trial Ethics

The following information was supplied relating to ethical approvals (i.e., approving body and any reference numbers):

The Brazilian Registry of Clinical Essays (ReBEC, Registro Brasileiro de Ensaios Clínicos –www.ensaiosclinicos.gov.br) approved this research. (Ethical Application Ref: RBR-7QNSH6) http://www.ensaiosclinicos.gov.br/rg/RBR-7qnsh6/.

## Data Availability

All raw data are provided in the Supplemental Files.

## Clinical Trial Registration

The following information was supplied regarding Clinical Trial registration:

rbr-7qnsh6.

## Supplemental Information

Supplemental information for this article can be found online at http://dx.doi.org/10.7717/peerj.6022#supplemental-information.

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
