# Peer review of "The effects of Pilates vs. aerobic training on cardiorespiratory fitness, isokinetic muscular strength, body composition, and functional tasks outcomes for individuals who are overweight/obese: a clinical trial"

_PeerJ, doi:10.7717/peerj.6022_

## Round 0.1 · original submission · Major Revisions

Dear authors,

Your manuscript has been reviewed by several experts in the analyzed topic and they have found scientific merit in your work. However, there are some issues which you must solve in a revised version of the text. Please, see the comments below in order to have more information.

With respect and warm regards,
Dr Palazón-Bru (academic editor for PeerJ)

·

Basic reporting

I would like to congratulate the authors for their effort invested in this article.
The article has minor English mistakes such as the use of “of” before Pilates in line 23 of the manuscript; line 113 the word “from” is wrongly placed twice. Also consider changing “be sedentary habits” to “had sedentary habits” in line 116. The authors should also be careful to always use past tenses. In lines 119/120 the English should be revised (maybe simplifying e.g., the participants performed a medical exam taking into account x and y).
The background is clear and objective, with valuable references that show the relevance of the study.
Although the structure of the manuscript follows a line of reasoning, lines 133/134 report results (mean ages) that from my point of view, should be placed only in the results section as part of the participants’ characteristics.

Experimental design

The research is within the scope of the journal and is well designed.
The authors brilliantly identified that there is a lack of reliable evidence on Pilates effectiveness in obesity or overweight patients and stated correctly that as previous studies have shown evidence is still needed about cardiorespiratory fitness outcomes.
As a suggestion I believe the article would benefit from a brief explanation (lines 192/193) of why the authors needed to monitor HR during the intervention. Was there any formula used to prescribe intensity? Was the theoretical maximum HR computed just to ensure a safe exercise?

Another great addition to line 203/204 would be an example of a possible adaptation of the exercises to an overweight or obese patient.

The protocol for the CPET was not clearly identified. From line 234 to 239 I encourage the authors to identify the original name of the protocol and the corresponding reference.

It is not clear for the reader why the authors needed 2 different measures for waist circumference. I think it would be helpful if you give a brief explanation.

Validity of the findings

The results are meaningful and their benefit to the literature is clear. However, lines 493-495 could be further explored. As the authors recognized, moderate exercise might not be enough to change body composition. However, it seems to change functional capacity. Why is that important? Is their a relation of functional capacity with other health outcomes in obese/overweight people? Is it a predictor of an important outcome? What are the implications of improving functional performance of obese patients?

In lines 524/525 – The authors should try to reformulate the sentence. Although walking is one of the easiest tasks it is one of the few where almost all musculoskeletal system is working. I think what you tried to state was that walking is not as demanding as Pilates for those specific muscular groups. If not, please try to provide a clearer statement.

In conclusions, it would be helpful to have a paragraph with the authors opinions regarding the need (or not) for future research, and if so, which specific aspects should be adressed.

Additional comments

I would like to thank the editor and the authors for the opportunity to review this work. It is a well-structured manuscript with interesting findings that cover an emergent topic of public health, and an intervention with potential for such population.

Reviewer 2 ·

Basic reporting

This article is self-contained, professionally presented and appropriately structured, and is comprehensively and thoroughly embedded in existing literature.

The article is generally well-written in clear and professional English. There are a few occasions where clarity could be improved or grammar is incorrect. These are mostly minor and I have provided some suggested changes below. I feel that the discussion could do with some more extensive rewriting, however.

1. I found the discussion quite long and somewhat convoluted. It would help to perhaps use subheadings for the various outcomes and sharpen the arguments by leaving out details already reported in the results and leaving out extraneous explanations. For example, the explanation in lines 466-471 is not really necessary, and the argument in lines 472-474 could be condensed to something like ‘Due to the small effect size, the results should be viewed with caution’.

2. Additionally, it would be helpful to start the discussion with a summary of the findings, outlining which variables (a) changed significantly over time for each group, and (b) were significantly different between groups at post-test.

3. Minor changes suggested:
Line 23: add ‘has’ before ‘for’
Line 67: change ‘to improve’ to ‘improved’
Line 75: on what specifically has there no consensus been reached?
Line 85: This sentence suggests adherence levels to traditional aerobic programs are not very high only if they are monotone and potentially painful for obese individuals. I wonder if this is what you mean to say (as it seems unlikely that these are the only circumstances where adherence levels are low); perhaps you can add ‘particularly’ if they are monotone…
Line 108/109: could you specify to which outcomes this hypothesis applies? Did you expect similar or higher benefits for all outcomes?
Line 113: delete first ‘from’
Line 120: change ‘participant first group’ to ‘first group of participants’
Line 133: change ‘that’ to ‘who’
Line 136: it is not clear what you mean by ‘recollections’
Line 152: add ‘to’ after ‘through’
Line 166-168: Please rephrase these sentences. People are blinded to one or more experimental conditions, they are not blinded in themselves (i.e., don’t use blinded as an adjective)
Line 167: change ‘have already’ to ‘had’
Line 183: insert a comma between ‘protocol’ and ‘for’
Line 207: should this be ‘ventilatory’?
Line 213: delete ‘then’
Line 326: change ‘was’ to ‘is’
Line 340: delete ‘of’ before ‘2%’
Line 452: add a comma after ‘Probably’
Line 456: change ‘once’ to ‘as’
Line 613: delete ‘in the baseline’

Experimental design

The study was well-designed and mostly described in excellent detail, sufficient to be reproducible by another investigator. There are a few issues that need further clarification, however:

4. Line 116: the criterion of ‘sedentary habits’ requires clarification. First, although the term ‘sedentary’ used to be common to refer to lack of sufficient physical activity, nowadays it tends to be reserved for actual sedentary behaviours (i.e., “any waking behaviour characterized by an energy expenditure ≤1.5 METs while in a sitting or reclining posture”, see Network SBR. Letter to the Editor: standardized use of the terms “sedentary” and “sedentary behaviours”. Appl Physiol Nutr Me. 2012;37:540–2. doi:10.1139/H2012-024). Second, if the criterion refers to lack of sufficient physical activity, it would be pertinent to know which level of physical activity is not met. I would thus recommend that this criterion is expressed in terms of inactive or insufficiently active in the sense of not meeting specified physical activity guidelines.

5. Lines 118 and 123: I may be missing something here but cardiovascular disease and arrhythmia do not appear to be part of the listed exclusion criteria, yet you have used them for exclusion.

6. Line 136-141: Could you specify when these comorbidities were detected? Was this during the screening exam? How do they relate to the exclusion criteria?

7. Lines 222-229: Please specify when the food intake assessments were conducted.

8. Line 352: you do not appear to have reported the results of these one-way ANOVAs. See also my comment on the possible use of ANCOVAs.

Validity of the findings

9. I am not a statistician but I believe the potential confounding effect of age differences between the Pilates group and the other groups could be statistically accounted for by conducting Analyses of Covariance rather than Analyses of Variance as the authors have done. I strongly recommend seeking a statistician’s advice on this issue.

Additional comments

I believe this is a good study and deserves publication after the recommended changes. My recommendation of major revision is mostly based on my concern about potential changes to the results when age is included as a covariate.

Reviewer 3 ·

Basic reporting

Too many definition-reference mismatches.

Experimental design

no comment

Validity of the findings

no comment

Additional comments

-Manuscript is too long, almost twenty pages or more. Manuscript should be written in fluent, clear and short form. Readers need to be able to easily read, interpret and understand article.
-Abstract- Method section: Please, indicate which evaluations were made.
- The literature/references should be reviewed again throughout the whole article. I have given a some examples of definition-reference mismatches below.
Line 76-78, "Moderate walking or jogging programs have traditionally been suggested to improve variables related to health (Ross & Janssen, 1999; Pedersen & Saltin, 2015; Santangelo et al., 2016)..." Santangelo et al 2016 Pathophysiology of obesity on knee joint homeostasis: contributions of the infrapatellar fat pad. -- this article related to development and progression of OA, and the contribution of fat pads. I could not find any comments about "moderate walking or jogging programs".
Line 78-81, "however, randomized controlled trials (RCT) suggest that walking (≥ 12 weeks) results in only a small beneficial effect on body mass, body composition, and aerobic fitness in overweight and obese adults (Gourlan, Trouilloud & Sarrazin, 2011; Aladro-Gonzalvo et al., 2012; Santangelo et al., 2016; Burgess et al., 2017)." Aladro-Gonzalvo et al. The effect of Pilates exercises on body composition: A systematic review.---this reference about Pilates but description about ≥ 12 weeks walking . Santangelo et al., I could not find any comments about "moderate walking or jogging programs".
- Exclusion criteria must be clearly stated. Line 117-118, Exclusion criteria were present neurological, cognitive, orthopedic, respiratory, and/or endocrine disease. Authors excluded 45 volunteer with cardiovascular disease and one volunteer with arrhythmia. Are cardiovascular disease and heart diseases within the criteria of exclusion? Line 136-141, diseases such as diabetes mellitus and hypothyroidism are endocrine diseases. but you indicated that endocrine diseases are in the exclusion criteria. This section is very confusing.
-Line 167-168, "Since previous studies investigating aerobic training effects have already found positive results with eight weeks of training (Ho et al., 2012; Many et al., 2013)..." Ho et al's study about vascular function; Many et al's study about insulin sensitivity and inflammatory markers in obese and insulin-resistant minority adolescents. Your study sample consist of "relatively healthy" adults. For this reason, the studies you have been referring to for the aerobics training protocol (8 week/3times per week...) does not compliance your aim. In addition, one of your outcome measures is body composition. The aerobic training load required to make a change in body composition is clearly indicated by the ACSM criteria (American College of Sports Medicine Position Stand. doi: 10.1249/MSS.0b013e3181949333 and Matthew A. Exercise Aspects of Obesity Treatment). Although Pilates performed 8 weeks in previous studies, aerobic training is usually performed for 12 weeks. From this point of view, the result seems to be determined at the beginning of the study.
-Line 207-211, authors stated that walking intensity in aerobic training protocol created according to Gourlan, Trouilloud & Sarrazin, 2011 and it did not lead to orthopedic injuries. In this article I cannot find no comment about orthopedic injuries and Gourlan et al stated that " Given these results and the fact that the concept of ‘dose’ of intervention has yet to be explored and developed for educational treatments, it seems that more research is needed in this domain, notably to determine whether or not there is an optimal number or frequency of sessions for a given duration of intervention." The aerobic training protocol should be clearly stated. In addition to this, study sample consist of sedentary overweight and obese. Guidelines usually recommend; instead of starting to walk 180 minutes a week walking time/distance be increased gradually. Please, convey your opinions about this issue.
-In assessment section, please just mention about how assessments are carried out. For example: Line 272-276 and 290-291 authors mention about reliability studies. It is more appropriate to address/discuss them in the discussion section.
-Line 295-296, Colyer et al., 2016 carried out their study on athletes. Line 485, "considering that DXA—a gold-standard method—was used in our study............" Are there any studies about the reliability of DXA and about whether DXA is considered gold standard in overweight and obese ?
-Line 304, abdominal endurance test; Is it sit-up or curl-up test?

---

## Round 0.2 · Minor Revisions

Dear authors,

Still pending some minor issues which you must address in a revised version of the text.

With respect and warm regards.
Dr Palazón-Bru (academic editor for PeerJ)

·

Basic reporting

I congratulate the authors for the improved version of the manuscript. Please try to be coherent through out the body of the manuscript and choose either the term "Pilates" or "Pilates training".
In line 117 please remove the extra dot.
In line 119 please replace "also were" with "were also".
In line 122 also replace the order of medical and physical.
In line 125 replace "presentation" with "presence" and "presence of arrhythmia...". Keep the format (n=x)

Line 172 Please replace the term "physical activity" with exercise, or "apart from the one included in this study."

Simplify line 368-369 (e.g., 90% confidence intervals and percent changes were also calculated).

In line 375 correct "significantly".
Line 429 replace "showed significantly improved" by "showed a significant improvement"

In some references there is no DOI. Try to maintain coherence.

The caption on figures is usually presented after the figure. See figure 1

Experimental design

The authors should consider moving the assessments up in the manuscript before the interventions, as this was the first step.

Validity of the findings

no comment

Additional comments

Overall you should try to shorten the manuscript as it becomes a little difficult for the reader. As a suggestion you may not emphasize as much the importance of effect sizes, as they are now well-recognized in the scientific community.

Reviewer 2 ·

Basic reporting

Thank you for including a summary of the findings at the beginning of the discussion. The discussion is still fairly long but the use of sub-headings has made the structure much clearer, and the arguments are more focused.

A few minor changes suggested:
(Please note that where I refer to line numbers, these are the line numbers in the word document with tracked changes [peerj-29176-manuscript_R1_final.docx].)

Line 157: You may want to spell out the full name of IPAQ in the paper for readers not familiar with this abbreviation.

Line 160: Please remove the commas before and after “which were not controlled”. (With the commas, the sentence reads as if all subjects presenting with cardiovascular or endocrine disease were excluded, because for all of them, the diseases were not controlled. Without the commas, the sentence will indicate that only those subjects for whom the diseases were not controlled were excluded, which is what you mean.)

Lines 211-214: The phrasing of “blinded professions” is still awkward. Please revise this to something like: “Professionals who carried out the tests and measurements were blinded to the treatment conditions.” Delete the subsequent sentence starting with “Volunteers…”

Lines 408; 424; 437-442: it would be more logical to report the age differences before noting that the analyses involved age as a covariate. Please consider moving lines 437-442 to a section before line 408. This could potentially be part of the “participants” section. Line 424 should also be part of the information that is moved to before line 408.

Line 531: change “circunference” to “circumference”.

Line 533: I would suggest to change “However” to “Conversely”.

Experimental design

Thank you for clarifying the criteria for categorizing physical activity level. The explanations about exclusion criteria and their link with comorbidities are much clearer now. Just note my comment about commas (in line 160) above.

Validity of the findings

I was pleased to see the authors consulted a statistician to get advice on running an Analysis of Covariance to control for the age differences between the Pilates and other groups. Having accounted for a possible age effect this way, the paragraph in the discussion (lines 714-719) probably does not need to be included.

Additional comments

The authors should be commended on their extensive and detailed response to the reviewers’ suggestions.

---

## Round 0.3 · Minor Revisions

Dear authors,

Still pending some very minor revisions in your paper. Please, see the comments of the reviewer.

With respect and warm regards,
Dr Palazón-Bru (academic editor for PeerJ)

Reviewer 2 ·

Basic reporting

I am happy with the revisions but I think there has been a confusion about which lines to move. I apologise if my earlier instructions were unclear, as it looks like you moved the wrong lines (409-410 rather than 411). (I am referring to the line numbers in the tracked changes document, with all revisions visible, i.e., manuscript_R2_final_tracked_changes.docx, with all markup visible)

To fix this, move lines 164-165 (When the age presented…. was used.) back to lines 409-410. They are more logical there. Perhaps change “When the age presented” with “Where age presented”. Then move line 411 (One-way ANOVA was used…. intervention.) to line 161, before “The one way ANOVA” (then add the explanation “analysis of variance” after the first mention of ANOVA).

Experimental design

no comments

Validity of the findings

no comments

Additional comments

Thank you for all your efforts in this study. Sorry to ask for one more minor revision.

---

## Round 0.4 · accepted · Accept

Dear authors,

I am pleased to inform you that your paper has been accepted for publication in PeerJ.

Congratulations!

With respect and kind regards,
Dr Palazón-Bru (academic editor for PeerJ)